# OpenBox: Annotate Any Bounding Boxes in 3D

**In-Jae Lee**[1]  **Mungyeom Kim**[1]  **Kwonyoung Ryu**[2]  **Pierre Musacchio**[1]  **Jaesik Park**[1]

[1]Seoul National University  [2]POSTECH

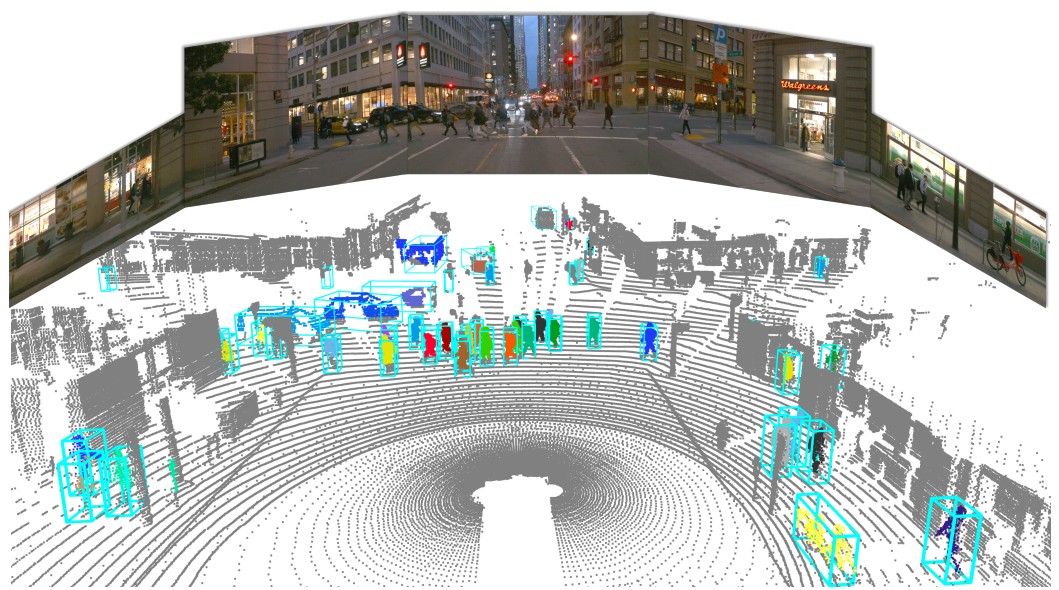

Figure 1: We introduce **OpenBox**, which utilizes a 2D vision foundation model to annotate 3D bounding boxes automatically. It annotates instances of vehicles, pedestrians, and cyclists. We demonstrate it with Waymo Open Dataset [33]. Best viewed in color and zoomed in.

## Abstract

Unsupervised and open-vocabulary 3D object detection have recently gained attention, particularly in autonomous driving, where reducing annotation costs and recognizing unseen objects are critical for both safety and scalability. However, most existing approaches uniformly annotate 3D bounding boxes, ignoring objects' physical states, and require multiple self-training iterations for annotation refinement, resulting in suboptimal quality and substantial computational overhead. To address these challenges, we propose **OpenBox**, a two-stage automatic annotation pipeline that leverages a 2D vision foundation model. In the first stage, OpenBox associates instance-level cues from 2D images processed by a vision foundation model with the corresponding 3D point clouds via cross-modal instance alignment. In the second stage, it categorizes instances by rigidity and motion state, then generates adaptive bounding boxes with class-specific size statistics. As a result, OpenBox produces high-quality 3D bounding box annotations without requiring self-training. Experiments on the Waymo Open Dataset (WOD), the Lyft Level 5 Perception dataset, and the nuScenes dataset demonstrate improved accuracy and efficiency over baselines. Our project page is available at: https://oliver0922.github.io/OpenBox/.

39th Conference on Neural Information Processing Systems (NeurIPS 2025).

# 1 Introduction

3D object detection has become increasingly important across a wide range of applications, including autonomous driving [16, 21, 23, 42], robotics [29, 41], and virtual reality [15, 37]. In autonomous driving, it provides essential inputs for motion prediction that, in turn, inform path planning and vehicle control. As a result, the accuracy of 3D object detection is directly tied to the overall safety and reliability of the system. While recent advances in deep learning have significantly improved detection performance, most existing frameworks [16, 21, 23, 42] remain limited to a fixed set of object categories and are heavily reliant on large-scale, human-annotated datasets. This closed-set assumption becomes particularly problematic in open-world autonomous driving scenarios. In such settings, the system must be able to detect a wide range of object types, including rare or previously unseen instances.

Integrating open-vocabulary detection enables models to recognize arbitrary categories based on semantic descriptions, thereby overcoming the limitations of a fixed label space. In the 2D image domain, open-vocabulary perception has been accelerated by the availability of large-scale image-text paired datasets and the emergence of vision foundation models. These models demonstrate strong generalization capabilities across tasks such as classification [30, 34], detection [22, 24, 48], and segmentation [14, 19, 31].

Despite the advances above, creating large-scale annotated 3D datasets remains a major bottleneck. Unlike 2D images, LiDAR point clouds provide precise geometric structure but lack rich semantic context, making them difficult to align with text-based supervision and challenging to annotate manually. To address these limitations, several unsupervised methods [40, 43, 45, 46] have been proposed. These typically follow a pipeline in which ground points are removed from raw LiDAR scans, spatial clustering is applied to extract object instances, and scene flow [1, 25, 26] or a persistence point score (PP Score) [40, 43, 46] is used to identify motion states. The resulting 3D bounding boxes are then refined through multiple rounds of self-training [43, 45, 46] or sampling strategies [46]. However, these methods generally do not consider physical properties of instances for box annotation, which leads to low-quality boxes and remains computationally expensive due to their iterative refinement. More recently, several works [18, 26, 46] incorporate image semantics to assist automatic annotation. Nevertheless, [46] fuses modality-specific 3D bounding boxes at the output level without geometric alignment, and [18] does not fully leverage visual cues to improve 3D annotation quality.

This paper proposes a two-stage pipeline, **OpenBox**, that automatically annotates 3D bounding box for arbitrary classes. Our approach leverages high-quality instance-level information from 2D vision foundation models (e.g., Grounding DINO [22], SAM2 [31]) as supervisory signals, thereby reducing the cost and time of manual annotation. In the first stage (Cross-modal Instance Alignment), 2D instance-level information is unprojected onto the 3D point clouds. To address noisy or incomplete instance point clouds caused by the vision foundation model, we apply a context-aware refinement step to enhance the quality of instance-level points. Subsequently, the refined instances are categorized into three physical types: static rigid, dynamic rigid, and deformable. Based on category-specific object size statistics, we generate 3D bounding boxes for each category. Specifically, we construct a mesh for static rigid objects using the Signed Distance Function (SDF) [36] and filter out noise points through majority voting. We then further refine the bounding box via 3D-2D IoU alignment and visibility. We conduct experiments on the WOD [33], Lyft [12], and nuScenes [2] datasets. Qualitative results on real-world data show that our method produces high-quality and robust 3D annotations, as illustrated in Fig. 1.

Our contributions are summarized as follows:

- We propose OpenBox, a novel automatic annotation pipeline that requires only synchronized ego poses, images, and LiDAR point clouds, without self-training.

- To improve point clouds quality, we introduce a two refinement process: context-aware refinement and surface-aware noise filtering based on the SDF. We also generate bounding boxes adaptively based on the physical types of instances.

- Training with OpenBox-generated annotations achieves **70.49**% $AP_{3D}$ for the vehicle class of the WOD [33] at 0.5 IoU. On the Lyft dataset [12], OpenBox outperforms the state-of-the-art approach by **+19.94**% $AP_{3D}$ when directly compared to human annotation boxes.

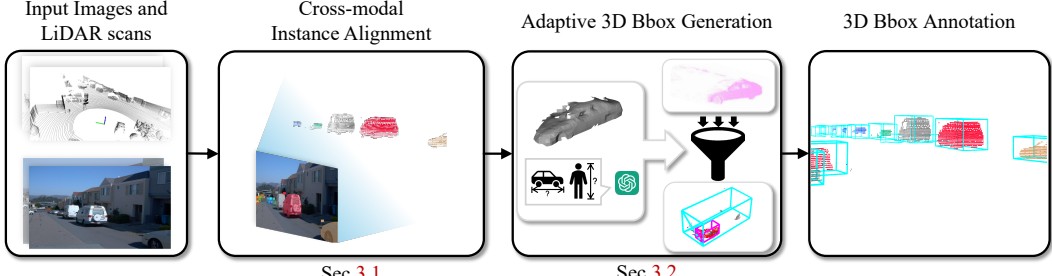

Figure 2: **Pipeline Overview of OpenBox.** With time-synchronized, unlabeled images and LiDAR scans, cross-modal instance alignment (Sec 3.1) associates 2D instance cues with corresponding point clouds. Adaptive 3D bounding box generation (Sec 3.2) independently chooses the most suitable fitting strategy for each instance, yielding high-quality 3D bounding boxes.

## 2  Related Work

### 2.1  Open-vocabulary 3D Object Detection

The rapid progress of 2D vision foundation models [11, 20, 22, 30, 34] has spurred active research in open-vocabulary 3D object detection. Most existing methods [8, 26, 44] focus on accurately aligning 2D visual cues with 3D spatial information. UP-VL [26] enhances the MI-UP [25] auto-labeling pipeline by incorporating OpenSeg [11] to generate semantically aligned amodal 3D bounding boxes for open-vocabulary transfer. Additionally, it introduces a loss function that facilitates 2D-3D mapping, allowing the model to learn point-level features guided by distillation loss. Find and Propagate [8] generates frustum-shaped 3D proposals using 2D open-vocabulary detectors [20, 24], followed by multi-view alignment and density-based filtering to improve the detection of distant objects. OpenSight [44] lifts 2D bounding boxes obtained from Grounding DINO [22] into 3D space to enable generic object perception followed by semantic interpretation. Whereas prior work redesign 3D object detectors for the open-vocabulary setting, we focus on annotating the dataset to allow open-vocabulary 3D detection.

### 2.2  Unsupervised 3D Object Detection

**LiDAR-based.** Most unsupervised 3D object detection methods [1, 25, 40, 43, 45] solely rely on LiDAR point clouds to perform automatic annotation. Common pipelines first estimate motion states using PP score [40, 43] or scene flow [1, 25], and then perform ground removal followed by point cloud clustering [3, 7]. Except for [40], which utilizes class-wise size statistics, these limitations primarily stem from the lack of semantic information inherent in LiDAR, especially when compared to RGB images. As a result, they generally train and evaluate models in a *class-agnostic* manner. Moreover, CPD [40] incurs additional computational overhead by jointly using dense prototypes (CProto) and down-sampled point clouds, resulting in significantly longer training time.

**Multi-modal based.** Several approaches [18, 26, 46] use 2D vision foundation models to incorporate image information. LiSe [46] integrates 3D bounding boxes obtained from the LiDAR branch (via [43]) and the image branch (via [14, 22, 38]) in a distance-aware manner. UNION [18] leverages appearance from 2D images to cluster and distinguish between mobile and immobile objects.

Unlike these approaches, which depend on iterative self-training [43, 46] and do not consider physical properties of instance [18, 26, 40, 43, 46], our method is designed to produce physical-state-specific annotations and to alleviate the need for iterative refinement.

## 3  Method

This section explains the role and design choices of each module in our proposed automatic 3D bounding box annotator, **OpenBox**. As shown in Fig. 2, our system consists of two main stages: (1) Cross-modal Instance Alignment and (2) Adaptive 3D Bounding Box Generation.

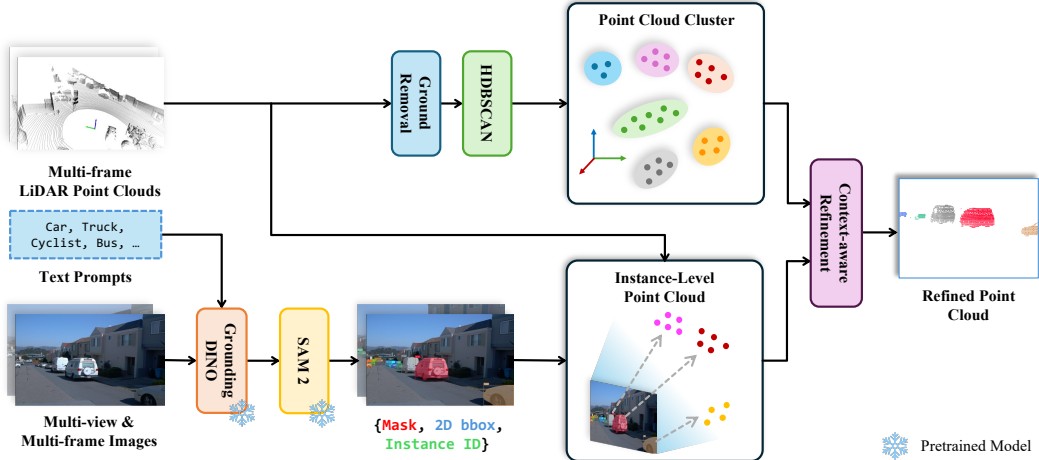

Figure 3: **Cross-modal Instance Alignment.** To obtain refined point clouds, the pipeline generates two complementary proposals. The LiDAR (upper) branch removes ground points and applies HDBSCAN [3] to produce coarse 3D clusters. The image-LiDAR (lower) branch uses Grounding DINO [22] followed by SAM2 [31] to generate 2D instance masks. This information is unprojected into point clouds. Context-aware refinement fuses the two proposals, discarding noisy points and incorporating adjacent points from these clusters, yielding refined per-object point clouds.

## 3.1 Cross-modal Instance Alignment

**Instance-level Feature Extraction.** As shown in Fig. 3, to leverage the strong capabilities of vision foundation models [22, 31] trained on large-scale datasets, we define a 3D-to-2D mapping function $\mathbf{\Pi}_j$ that projects the 3D point cloud $\mathcal{P}^{(t)} \in \mathbb{R}^{M \times 3}$ captured at time $t$ onto the $j$-th camera image $I_j^{(t)} \in \mathbb{R}^{3 \times H \times W}$. Using a 2D detector [22] $\Psi$ and a segmentation model [31] $\Phi$, we obtain 2D boxes $\mathcal{B}_j^{(t)}$, class labels $\mathcal{C}_j^{(t)}$, masks $\mathcal{M}_j^{(t)}$, and instance IDs $\mathcal{T}_j^{(t)}$ from $I_j^{(t)}$ as follows:

$$\mathcal{B}_j^{(t)}, \mathcal{C}_j^{(t)} = \Psi(I_j^{(t)}), \quad \mathcal{M}_j^{(t)}, \mathcal{T}_j^{(t)} = \Phi(\mathcal{B}_j^{(t)}, txt), \quad \mathcal{V}_j^{(t)} = \mathbf{\Pi}_j(\mathcal{P}^{(t)}, I_j^{(t)}), \qquad (1)$$

where $H$ and $W$ denote the image height and width, $\mathcal{B}_j^{(t)}$ the 2D bounding boxes, $\mathcal{V}_j^{(t)}$ the pixel coordinates of $\mathcal{P}^{(t)}$ projected onto $I_j^{(t)}$, and $txt$ the text prompts. By associating each projected 3D point with its corresponding image pixel and mask label, we obtain the instance-level point clouds $\mathcal{F}_j^{(t)} = \{\mathcal{F}_{ij}^{(t)} \in \mathbb{R}^{M' \times 6}\}_i$. Here, $\mathcal{F}_{ij}^{(t)}$ is the $i$-th instance-level point cloud, which contains 3D coordinates $(x, y, z)$, semantic class, instance presence, and instance ID. However, the boundaries of the masks obtained from [31] are imprecise, due to calibration errors, so directly unprojecting them into 3D can result in noisy point clouds. To mitigate this issue, we adopt the adaptive erosion proposed in [13], which erodes masks based on object size to eliminate boundary noise while preserving instance structure. For convenience, we omit the subscripts $t$ and $j$ from this point onward.

**Context-aware Refinement.** As shown in Fig. 4-(c), LiDAR points are often projected onto background regions (*e.g.* guardrail and wall) that occlude the actual foreground instance, resulting in inaccurate unprojection. These noisy points, located outside the true object region, tend to yield 3D bounding boxes that are improperly scaled. To address this issue, we refine the unprojected instance-level point clouds $\mathcal{F}$. We perform majority voting within clustered regions $\{\mathcal{R}_1, \mathcal{R}_2, \ldots, \mathcal{R}_{N'}\}$, where each cluster $\mathcal{R}_k \in \mathbb{R}^{m_k \times 3}$ is obtained from the ground-removed raw LiDAR point clouds $\mathcal{P}$ using HDBSCAN [3], following ground removal based on [17]. For each segment $\mathcal{R}_k$, we compare it with all instance-level point clouds $\mathcal{F}_i$ and compute bidirectional proximity-based inclusion ratios. Specifically, we determine the proportion of points in $\mathcal{R}_k$ that overlap with any point in $\mathcal{F}_i$, and vice versa. If mutual overlap between the two clusters is sufficient, the cluster $\mathcal{R}_k$ is assigned the instance ID $i$ and retained; otherwise, it is discarded. This process can be formulated as follows:

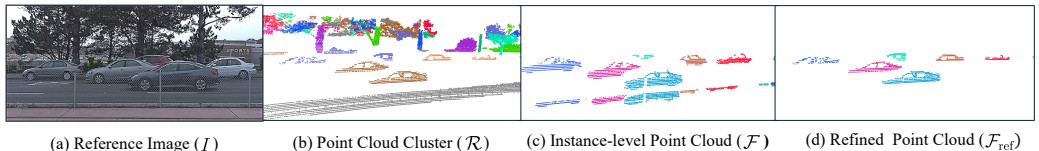

| (a) Reference Image ($I$) | (b) Point Cloud Cluster ($\mathcal{R}$) | (c) Instance-level Point Cloud ($\mathcal{F}$) | (d) Refined Point Cloud ($\mathcal{F}_{\text{ref}}$) |
|---|---|---|---|

Figure 4: **Context-aware Refinement.** (a) Reference image. (b) Point cloud clusters $\mathcal{R}$ after using HDBSCAN [3] on ground-removed LiDAR point clouds. (c) Noisy instance-level point clouds $\mathcal{F}$. (d) Result of the Context-aware refinement $\mathcal{F}_{\text{ref}}$.

$$\frac{|\{p \in \mathcal{R}_k \mid \text{dist}(p, \mathcal{F}_i) < \delta\}|}{|\mathcal{R}_k|} > \alpha, \quad \frac{|\{p \in \mathcal{R}_k \mid \text{dist}(p, \mathcal{F}_i) < \delta\}|}{|\mathcal{F}_i|} > \beta \quad \Rightarrow \quad \mathcal{R}_k \leftarrow i, \quad (2)$$

where $|\cdot|$ denotes the cardinality of a set, and $\text{dist}(p, \mathcal{F}_i) < \delta$ holds if and only if there exists $f \in \mathcal{F}_i$ such that $\|p - f\|_2 < \delta$.

## 3.2 Adaptive 3D Bounding Box Generation

Most prior methods [13, 43, 46] generate boxes without considering the physical properties of individual objects. This often leads to inaccurate localization and reduces the consistency of the data used to train 3D object detection networks. To address this issue, we propose an adaptive box generation strategy that accounts for the physical types of each instance.

**Static & Dynamic Points Decomposition.** The refined instance-level LiDAR point clouds $\mathcal{F}_{\text{ref}}$ obtained in the previous step remain sparse. Yet, by aggregating consecutive point cloud frames to a global coordinate system, the instance-level point clouds can be significantly densified. Incorporating point clouds from dynamic objects is challenging, as this may introduce motion artifacts [40]. Thus, we use the PP score [43] to estimate the ephemerality of each point in the refined point cloud.

**Initial Bounding Box Generation.** Empirically, we found that categorizing each instance based on its physical properties leads to better performance. In particular, we divide instances $\mathcal{F}_{\text{ref}}$ into three types: rigid and static $\mathcal{F}_{\text{ref}}^{S}$, rigid and dynamic $\mathcal{F}_{\text{ref}}^{D}$, and deformable $\mathcal{F}_{\text{ref}}^{deform}$. For each type, we generate a corresponding 3D bounding box. We use ChatGPT [27] to determine the object type based on the given semantic class to distinguish between rigid and deformable objects. Then, using this classification in conjunction with motion cues estimated via the PP score [43], we generate an appropriate 3D bounding box for each instance. We initially generate a bounding box for all three object types using an approach [47] that maximizes the closeness of points to edges. However, due to the sparsity of the point clouds and occlusion, the resulting bounding box may underestimate the actual object size. To address this, we use ChatGPT [27] to retrieve the typical size of the object class in terms of length, width, and height. If any of the initial bounding box dimensions are smaller than 80% of the typical size, we adjust the box size as described in the following sections.

**Handling Static & Rigid Instances.** Although we densify the aggregated static instance point cloud $\mathcal{F}_{\text{ref}}^{S}$, it still contains noise due to limitations of the context-aware refinement. To suppress it, we apply a surface-aware filtering method based on proximity voting over mesh vertices. Specifically, we reconstruct a mesh surface $\mathbf{S}$ from the point clouds using SDF [36]. For each vertex $v \in \mathbf{S}$, we identify nearby foreground and background points using the proximity function $\mathcal{P}_C(\cdot, v)$ defined in Eq. 3, where $\tau$ denotes the distance between the point and the mesh vertex. We retain vertices where foreground associations dominate, forming the refined surface $\mathbf{S}_{\text{ref}}$ as defined in Eq. 4. The final refined static point cloud $\mathcal{F}_{\text{ref}}^{S,(2)}$ is then constructed by collecting all foreground points near the filtered surface vertices.

$$\mathcal{P}_C(\mathcal{F}, v) = \{p \in \mathcal{F} \mid \|p - v\|_2 < \tau\}, \tag{3}$$

$$\mathbf{S}_{\text{ref}} = \left\{v \in \mathbf{S} \mid |\mathcal{P}_C(\mathcal{F}_{\text{ref}}^{S}, v)| > |\mathcal{P}_C(\mathcal{F}_{\text{ref}}^{C}, v)|\right\}, \quad \mathcal{F}_{\text{ref}}^{S,(2)} = \bigcup_{v \in \mathbf{S}_{\text{ref}}} \mathcal{P}_C(\mathcal{F}_{\text{ref}}^{S}, v). \tag{4}$$

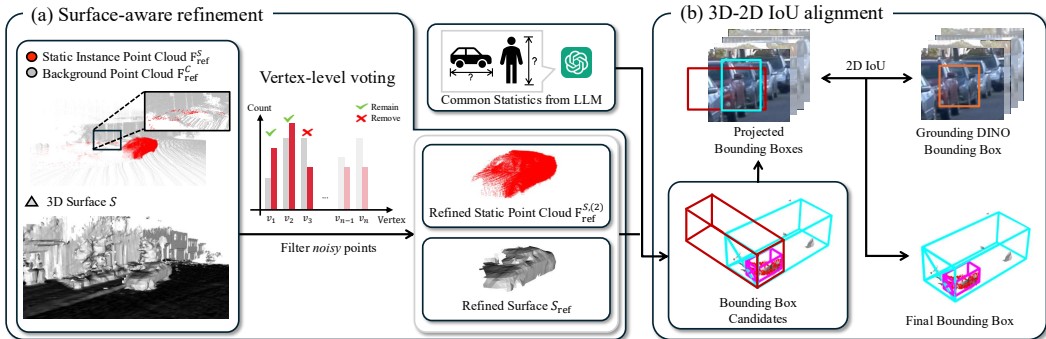

Figure 5: **Handling Static & Rigid Instances.** (a) We filter noisy points in the aggregated static point clouds via vertex-level voting on the reconstructed surface, producing $\mathcal{F}_{ref}^{S,(2)}$ and $\mathbf{S}_{ref}$. (b) We then adjust the bounding box using surface normals and statistical priors, and select the final box based on 2D IoU between projected boxes and Grounding DINO [22] boxes.

To refine the initial bounding box, we extract the corresponding instance-level surface mesh $\mathbf{S}_{ins}$ from $\mathbf{S}$. As shown in Fig. 5, if the box is too small, we determine the resizing direction using surface normal vectors, rather than searching over 8 directions as in [13]. We rotate the surface into the ego coordinate system. Then, we compute the dot product between surface normals and the four orthonormal reference vectors to determine which sides of the object are represented. If all four sides are covered, no resizing is needed. Otherwise, we generate two resized box candidates based on statistical priors. This is because the longer side of the initial box cannot be reliably assumed to represent the object's actual length. (Please see Sec. C.3 for more details.) To select the optimal box, we match each 3D candidate with 2D bounding boxes (from Sec. 3.1) using instance ID, project them onto multiple views and time steps, and compute their 2D IoUs. The box with the higher IoU is selected as the final result.

**Handling Dynamic & Rigid Instances.** In these cases, we rely on point clouds from a single moment in time, which makes it harder to accurately estimate the object's position, orientation, and size. To deal with this problem, OpenBox uses the fact that the orientation of a dynamic object is approximately aligned with the direction of the position difference in adjacent frames. OpenBox first estimates the object's orientation by the direction of the object trajectory associated with 2D tracking IDs. We then rotate the initial bounding box to align it with the estimated orientation angle. After aligning the orientation, we refine the box size. For each face along the X and Y axes, we compute the dot product between the outward surface normal and the LiDAR ray direction at the face center. As shown in Fig. 6, OpenBox extends the box only when the dot product between the ray and face normal is negative. We determine the final box size using standard object-size statistics.

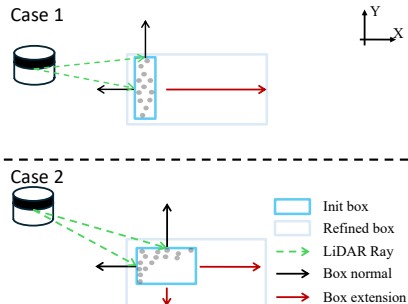

Figure 6: **Visibility-based box extension.** Case 1 has one negative value from the dot product between the ray and the normal, yielding a one-sided extension, whereas Case 2 has two negative values, leading to a two-sided extension.

**Handling Deformable Instances.** Deformable instances such as pedestrians, animals, or cyclists exhibit articulated or non-rigid motion, causing spatial inconsistencies across frames that often lead to ghosting or distorted geometry when aggregated [4]. Due to their limited surface structure, geometry-based refinement is ineffective. To maintain reliability, we generate bounding boxes from a single frame by tightly fitting the visible region using the closeness-to-edge algorithm [47], which provides robust representations without relying on rigid geometric assumptions.

Table 1: **3D object-detection results on the WOD [33] validation set.** * indicates trained and evaluated in the camera-frustum region, while others use full 360° coverage. † and ‡ denote models trained with CST and CBR from CPD [40], using the training settings given in the next sentence. For †, we flip the **OpenBox** annotations and point clouds to obtain 360° coverage; for ‡, we fill the region outside the camera frustum with CPD annotations. All values denote $AP_{3D}$ at each IoU threshold. **Bold** means best performance, underlined means second-best. Only L1 results are shown here; we provide the full L2 results in the Table 7.

| Method | Modality | Vehicle $IoU_{0.5}$ / $IoU_{0.7}$ | Pedestrian $IoU_{0.3}$ / $IoU_{0.5}$ | Cyclist $IoU_{0.3}$ / $IoU_{0.5}$ |
|---|---|---|---|---|
| CPD* [40] | LiDAR | 30.30 / 20.90 | 14.28 / 11.22 | 3.47 / **3.08** |
| **OpenBox* (Ours)** | LiDAR + Camera | **70.49 / 32.41** | **57.95 / 17.11** | **20.81** / 2.15 |
| DBSCAN [7] | LiDAR | 2.32 / 0.29 | 0.51 / 0.00 | 0.28 / 0.03 |
| MODEST [43] | LiDAR | 18.51 / 6.46 | 11.83 / 0.17 | 1.47 / 1.14 |
| OYSTER [45] | LiDAR | 30.48 / 14.66 | 4.33 / 0.18 | 1.27 / 0.33 |
| CPD [40] | LiDAR | 57.79 / 37.40 | 21.91 / 16.31 | 5.83 / 5.06 |
| **OpenBox† (Ours)** | LiDAR + Camera | **66.89** / 39.14 | **55.71 / 37.82** | **21.00 / 7.08** |
| **OpenBox‡ (Ours)** | LiDAR + Camera | 59.09 / **40.68** | 39.09 / 28.16 | 8.27 / 6.23 |
| Human Annotation | - | 93.31 / 75.70 | 87.25 / 77.93 | 58.84 / 54.88 |

Table 2: **3D object-detection results on the Lyft [12] validation set.** Following [46], we evaluate in class-agnostic manner at IoU = 0.25, and each value represents $AP_{BEV}$ / $AP_{3D}$. **Bold** means best performance, underlined means second-best.

| Method | Modality | 0–30m | 30–50m | 50–80m | 0–80m |
|---|---|---|---|---|---|
| MODEST-PP ($T=0$) [43] | LiDAR | 46.4 / 45.4 | 16.5 / 10.8 | 0.9 / 0.4 | 21.8 / 18.0 |
| LiSe ($T=0$) [46] | LiDAR + Camera | 54.5 / 54.0 | 24.2 / 22.8 | 1.4 / 1.2 | 29.2 / 27.5 |
| **OpenBox (Ours)** | LiDAR + Camera | **62.4 / 62.3** | **56.6 / 50.6** | **19.9 / 19.5** | **49.6 / 43.3** |
| Human Annotation | - | 82.8 / 82.6 | 70.8 / 70.3 | 50.2 / 49.6 | 69.5 / 69.1 |

## 4 Experiments

### 4.1 Experimental Setup

**Dataset and Implementation Details.** We conduct experiments on Waymo Open Dataset (WOD) [33], Lyft Level 5 Perception Dataset (Lyft) [12], and nuScenes [2]. For 3D object detection networks, we train Voxel R-CNN [6] for WOD [33], PointRCNN [32] for Lyft [12] and CenterPoint [42] for nuScenes [2] following the baselines [18, 40, 43, 46]. We refer readers to WOD [33], Lyft [12] and nuScenes [2] for details of the evaluation metrics. Our code is based on OpenPCDet [35] and MMDetection3D [5]. Additional details on training, hyperparameters, and network architecture are provided in Appendix B.

**Baselines.** In the WOD [33] benchmark, the state-of-the-art method CPD [40] evaluates the reliability of 3D bounding boxes using the CSS score and constrains network training by jointly learning dense CProtos within those boxes. For the Lyft [12] dataset, LiSe [46] fuses 3D bounding boxes from an image branch [38] and a LiDAR branch [43] based on distance. Finally, for nuScenes [2], UNION distinguishes mobile objects by leveraging visual appearance features extracted with DINOv2 [28]. Since our method in WOD [33] performs annotation only on point clouds that fall within the camera frustum field of view (FOV), we conduct experiments under two different settings.

**Experimental Scenarios.** We conduct experiments under two scenarios to comprehensively evaluate the quality of our automatic annotations. *Scenario 1* trains a 3D object detector on automatically annotated data and evaluates it on a human-annotated validation dataset. *Scenario 2* directly compares the automatic annotations with the human annotations on the training set.

Table 3: **Annotation performance on Lyft [12] training dataset.** We evaluate our automatically annotated dataset with a human-annotated dataset. Following [46], we evaluate in class-agnostic manner at IoU = 0.25, and each value represents $AP_{3D}$. **Bold** means best performance.

| Method | 0–30m | 30–50m | 50–80m | 0–80m |
|---|---|---|---|---|
| LiSe [46] | 17.47 | 6.87 | 1.35 | 6.31 |
| **OpenBox (Ours)** | **56.62** | **28.10** | **6.47** | **26.25** |

Table 4: **3D object-detection results on the nuScenes [2] validation set.** Following [18], we evaluate for 3 classes, and each value represents $AP_{3D}$. **Bold** means best performance.

| Method | Modality | Car | Pedestrian | Cyclist |
|---|---|---|---|---|
| UNION [18] | LiDAR + Camera | 30.1 | 41.6 | 0.0 |
| **OpenBox (Ours)** | LiDAR + Camera | **40.9** | **62.7** | **5.2** |

## 4.2 Main results

**Comparison on WOD.** Table 1 presents the LEVEL_1 $AP_{3D}$ results of experiments conducted on the WOD [33] under *Scenario 1*. For a fair comparison with the state-of-the-art method CPD [40], we conduct the experiments under two FOV (Field of View) settings. Under the camera-frustum FOV setting, our method consistently outperforms CPD [40] for vehicle and pedestrian classes, even though CPD incorporates additional training techniques (e.g., CST and CBR) beyond its annotation pipeline. The inferior performance of the cyclist class can be attributed to our use of the prompt "bicycle" in Grounding DINO [22], which often yields undersized bounding boxes compared to those enclosing the entire cyclist. Furthermore, we evaluate two extended settings: (1) applying CPD's [40] training schemes (CST and CBR) to our boxes, and (2) combining our boxes with CPD's [40] and then training with CST and CBR. Both approaches lead to performance improvements across all classes, with particularly notable gains for pedestrian and cyclist categories. We attribute this to a fundamental design difference: CPD [40] annotates only stationary objects, resulting in low recall. Furthermore, it relies on class-agnostic tracking and classifies based on box-size statistics. In contrast, our method identifies the object class using a 2D vision foundation model [22], and generates adaptive bounding boxes that reflect each class's physical properties, resulting in more accurate annotations.

**Comparison on Lyft.** Table 2 shows the results of class-agnostic 3D object detection on the Lyft [12] dataset under *Scenario 1*, using an IoU threshold of 0.25. To ensure a fair comparison, we evaluate against baseline methods [43, 46] that do not assume multiple traversals and do not apply self-training strategies. Our method demonstrates improved performance in both $AP_{BEV}$ and $AP_{3D}$ across all distance ranges compared to baselines. In particular, for long-range scenarios (50–80m), our method outperforms LiSe [46] by +18.5% in $AP_{BEV}$ and +18.4% in $AP_{3D}$. Furthermore, as shown in Table 3, we evaluate the performance of automatic annotations in the *Scenario 2* environment. Our method consistently outperforms LiSe [46] across all ranges. This performance gap arises because LiSe [46] integrates 3D boxes from the image branch, generated using the method of [38], and from the LiDAR branch, based on [43]. However, neither of these components explicitly considers the physical properties or semantics of the object classes, which limits their precision.

**Comparison on nuScenes.** As shown in Table 4, we observe performance improvements across all classes under *Scenario 1*. One key reason for the significant gains is that, unlike OpenBox, UNION [18] omits the refinement process for point clouds and 3D bounding boxes. In particular, it does not explicitly consider the camera-lidar calibration error when projecting LiDAR point clouds on DINOv2 [28] feature maps which leads to noise at the boundary of the objects. Additionally, UNION [18] neither resizes nor relocalizes the initial 3D bounding boxes, leading the model to predict suboptimal bounding boxes.

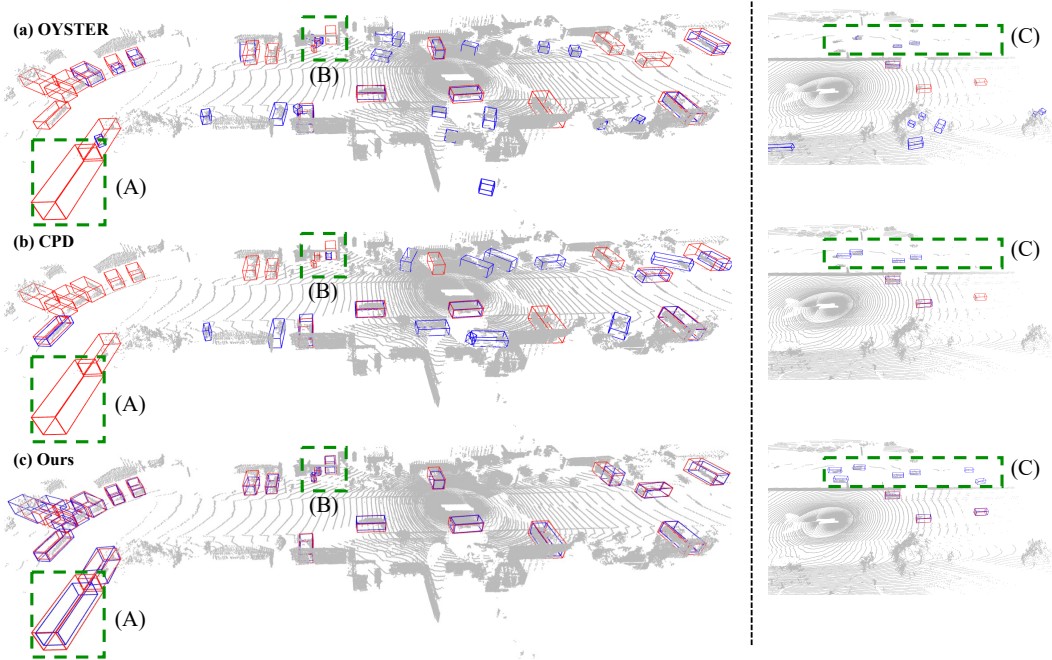

Figure 7: **Comparison of automatic annotation on WOD [33] training set.** Each row compares automatically annotated boxes with human-annotated boxes, while each column corresponds to a different scene. Blue boxes represent the automatically generated boxes, and red boxes indicate the human-annotated boxes. We visualize CPD [40] annotations filtered by a minimum CSS score threshold. Best viewed in color and zoomed in.

Table 5: **Ablation study on the *Vehicle* class for WOD [33] training set.** Surface-aware, Context-aware, and 3D-2D IoU refer to Surface-aware Refinement, Context-aware Refinement, and 3D-2D IoU alignment respectively. All results stand for the Vehicle class using $AP_{3D}$ at IoU = 0.4.

| (a) Point-level refinement | | | | (b) Box-level refinement | | |
|---|---|---|---|---|---|---|
| Surface-aware | Context-aware | $AP_{3D}$ | | Visibility-based | 3D-2D IoU | $AP_{3D}$ |
| ✓ | | 30.34 | | ✓ | | 30.49 |
| | ✓ | 32.52 | | | ✓ | 34.71 |
| ✓ | ✓ | **38.65** | | ✓ | ✓ | **38.65** |

## 4.3 Ablation study

To analyze the impact of each module on automatic annotation, we conduct an ablation study under the *Scenario 2*. In Table 5-(a), we observe that applying both point-level refinement modules yields the highest performance. This is because Context-aware Refinement is applied to all instances regardless of their physical properties, whereas Surface-aware Refinement is specifically designed for rigid and static instances. Furthermore, some effects of Surface-aware Refinement are partially covered by Context-aware Refinement, which explains its relatively larger contribution when applied alone. Similarly, Table 5-(b) presents an ablation study focusing on modules that contribute to box-level refinement. The 3D-2D IoU alignment module is designed to resize and relocate boxes for static and rigid instances, while the visibility-based module is applied to dynamic and rigid instances. 3D-2D IoU alignment has a greater overall impact when combined with the visibility-based method for two main reasons: (1) the number of static vehicles in the WOD [33] dataset is significantly larger than that of dynamic vehicles, and (2) the two candidate boxes considered by 3D-2D IoU alignment typically differ by 90°, leading to a more substantial effect on the $AP_{3D}$.

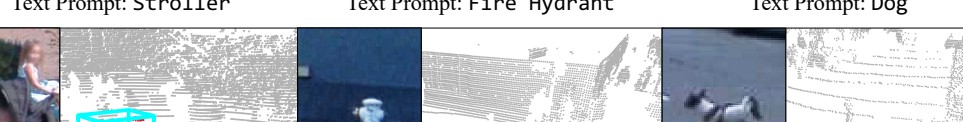

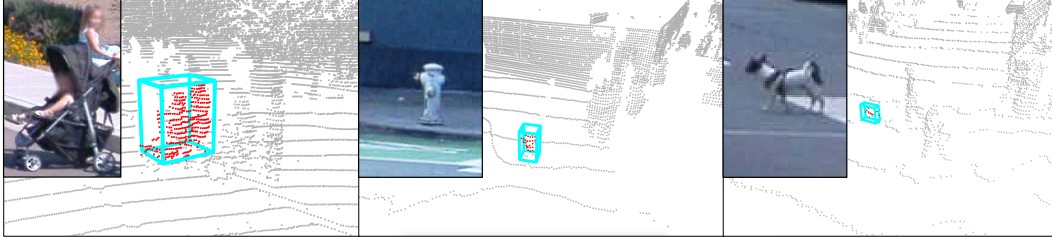

Figure 8: **Our automatic annotation results on novel classes in WOD [33].** In this visualization, red points represent instance-level point clouds, while cyan boxes indicate the automatically generated annotations. Best viewed in color and zoomed in.

## 4.4 Qualitative Result

As shown in Fig. 7, we compare 3D bounding boxes from automatic and human annotations. Overall, OpenBox shows higher precision and recall compared to the baselines [40, 45]. Region (A) illustrates a static travel trailer. Since [40, 45] generate boxes without considering an instance's physical properties, it remains unannotated despite being static. In contrast, OpenBox recognizes it as static, enabling annotation even with sparse point evidence. In region (B), our approach successfully detects a static car and a pedestrian inside a garage, which the baselines miss or mislocalize. This is because our method refines the point cloud to isolate instance-specific points. Region (C) shows that our method can automatically annotate vehicles on the opposite lane, even when no corresponding human annotations exist. Although both the baseline and our method detect these vehicles, ours localizes them more accurately by extending to both rigid and dynamic instances and by jointly leveraging a vision foundation model, resulting in higher recall. In addition, OpenBox enables automatic annotation of open-vocabulary objects beyond the predefined classes in existing autonomous-driving datasets [2, 10, 12, 33, 39]. As shown in Fig. 8, it successfully annotates objects such as strollers, fire hydrants, and dogs, which are essential to consider in real-world driving scenarios.

## 5 Conclusion

In this paper, we propose OpenBox, a novel automatic 3D bounding box annotation pipeline. Open-Box effectively leverages 2D vision foundation models to generate open-vocabulary 3D annotations. To ensure high-quality box generation, it refines instance-level point clouds and performs adaptive 3D bounding-box generation tailored to the physical properties of each instance. Our extensive experiments on diverse autonomous driving datasets demonstrates the superiority of the proposed method in annotation quality over prior baselines, while our comprehensive ablation study substantiates the effectiveness of each component in the annotation pipeline. We hope that our method can contribute to future research on foundation models for 3D perception.

**Limitations.** Adverse weather reduces contrast and obscures edges, which makes 2D vision models unreliable. The 3D annotations built on those models inherit the errors and often miss instances, resulting in imprecise box boundaries. Deformable categories such as pedestrians and cyclists also suffer because pose variation makes full-extent inference unstable. The method then falls back to fixed class-level sizes, which frequently under- or over-size the box. At long range, LiDAR returns become too sparse to constrain geometry, so box fitting is ill-conditioned and localization becomes less precise.

## 6 Acknowledgments

This work was supported by IITP grant (RS-2021-II211343: AI Graduate School Program (Seoul National University) (5%), No.2021-0-02068: AI Innovation Hub (10%), 25-InnoCORE-01: InnoCORE program of the Ministry of Science and ICT (10%)) and NRF grant (2023R1A1C200781211 (75%)) funded by the Korea government (MSIT).

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

# Appendix

## A  Overview

In this supplementary material, we provide additional details of OpenBox. Appendix B describes the experiment details. Appendix C provides a detailed approach for height refinement and surface estimation. We further present more experimental results in  Appendix D.

## B  Implementation Details

Table 6: Training and network details for experiment

| configs | Voxel R-CNN [6] | Point RCNN [32] | CenterPoint [42] |
|---|---|---|---|
| optimizer | AdamW | AdamW | AdamW |
| base learning rate | 1e-2 | 1e-2 | 1e-4 |
| weight decay | 1e-3 | 1e-2 | 1e-2 |
| momentum | 0.9 | 1e-2 | — |
| momentum range | [0.95, 0.85] | [0.95, 0.85] | — |
| learning rate decay | 0.1 | 0.1 | — |
| learning rate clip | 1e-7 | 1e-7 | — |
| gradient norm clip | 10 | 10 | 35 |
| batch size | 16 | 2 | 32 |
| epoch | 20 | 60 | 20 |

We train models [6, 32, 42] on 8 NVIDIA A6000 GPUs (48GB) and 2 AMD EPYC 7763 CPUs. We also employ VDBFusion [36] for SDF. In the Context-aware Refinement part (see Sec. 3.1), the hyperparameters $\alpha$, $\beta$, and $\delta$ are set to 0.3, 0.2, and 0.1, respectively. In addition, the threshold $\tau$ used in the Handling Static & Rigid Instance part (see Sec. 3.2) is set to 0.15.

## C  Additional details for method

### C.1  ChatGPT prompt

To obtain 3D bounding box size priors and determine the rigidity of objects, we utilized the following prompts with ChatGPT-4 [27]:

> **Prompt**:  Please provide the typical 3D bounding box size for
> [class].
>
> **Response**:  Here are the typical 3D bounding box dimensions
> (Length $\times$ Width $\times$ Height in meters) for [class], based on common
> datasets like nuScenes, KITTI, and the Waymo Open Dataset.
>
> **Prompt**:  Is [class] deformable or rigid?
>
> **Response**:  A [class] is considered a deformable / rigid object.

Below is the list of categories we provided to GroundingDINO [22] as text prompts:

> [Car, Bus, Person, Truck, Construction Vehicle, Trailer,
> Barrier, Bicycle, Motorcycle, Traffic Cone, Dog, Fire Hydrant,
> Stroller]

### C.2  Height Refinement

In section 3.1, we exclude points whose $z$-coordinates are below a predefined threshold to remove remaining ground points after the RANSAC [9] based ground removal. This preliminary step can lead

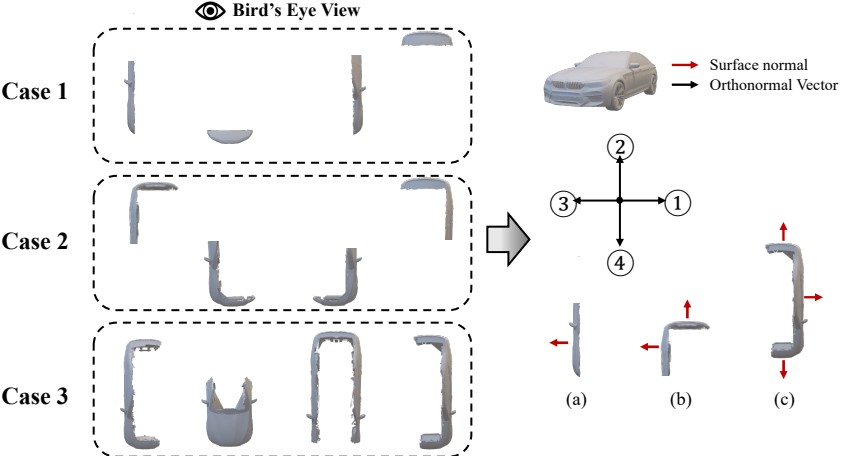

Figure 9: **Illustration of Surface Estimation.**

to bounding boxes appearing elevated above the actual object. To address this issue, we propose an additional refinement step to accurately estimate the $z$-coordinate of each bounding box. Specifically, given an instance with length $l$ and width $w$, we calculate a radius as:

$$L_b = \frac{\sqrt{l^2 + w^2}}{2}. \tag{5}$$

We then define the set of points within this radius from the ego-position $p_{\text{ego}}$ of the instance:

$$\mathcal{P}' = \{p \in \mathcal{P} \mid |p_{\text{ego}} - p|_2 < L_b\}, \tag{6}$$

where $\mathcal{P}$ denotes the point cloud corresponding to the frame in which the instance is located. After sorting points in $\mathcal{P}'$ by their z-coordinate in ascending order, we select the z-coordinate at the 1%, effectively removing potential noise and LiDAR reflectance outliers near the ground. This procedure ensures a robust estimation of the ground level near the instance.

## C.3 Surface Estimation

We determine the surface direction of instance-level surfaces $\mathbf{S}_{\text{ins}}$ to facilitate 3D-2D IoU alignment, as described in the **Handling Static & Rigid Instance** section (Sec. 3.2). Specifically, we compute the dot product between the normal vectors of $\mathbf{S}_{\text{ins}}$ and a set of four orthonormal reference vectors to identify the surface direction.

As illustrated in Fig. 9, in (a), only the ③ direction yields a dot product greater than the predefined threshold $\gamma = 0.8$, allowing us to identify the surface direction. In (b), both ② and ③ exceed $\gamma$, while in (c), ①, ②, and ④ all surpass the threshold, indicating the presence of multiple surface orientations.

## D  More Experimental Results

**Quantitative Results**  Table 7 presents the $\text{AP}_{3D}$ results under the LEVEL_2 of the WOD [33] validation split. Compared to the results under the LEVEL_1 criterion shown in Table 1, which reflects performance in easier scenarios, the overall performance is lower. Nevertheless, our approach outperforms other baselines [7, 40, 43, 45], indicating that the proposed dataset annotations enable the 3D object detection network [6] to learn effectively even under more challenging conditions.

**Qualitative Results**  We demonstrate OpenBox on the WOD [33] dataset in two scenarios. Scenario 1 compares annotation quality on the original WOD training set for vehicle, pedestrian, and cyclist classes. Scenario 2 presents annotation results for novel object categories. A detailed visualization of both scenarios is provided in the attached video (supple_video.mp4).

Table 7: **3D object-detection results on the WOD [33] validation set.** Models marked with * are trained and evaluated in the camera-frustum region, while others use full 360° coverage. † and ‡ denote models trained with CST and CBR from CPD [40], using the training settings given in the next sentence. For †, we flip the **OpenBox** annotations and point clouds to obtain 360° coverage; for ‡, we fill the region outside the camera frustum with CPD annotations. All values denote $AP_{3D}$ at each IoU threshold for LEVEL_2. **Bold** means best performance, underlined means second-best.

| Method | Modality | Vehicle $IoU_{0.5}$ / $IoU_{0.7}$ | Pedestrian $IoU_{0.3}$ / $IoU_{0.5}$ | Cyclist $IoU_{0.3}$ / $IoU_{0.5}$ |
|---|---|---|---|---|
| CPD* [40] | LiDAR | 26.09 / 17.91 | 11.87 / 9.30 | 3.34 / **2.96** |
| **OpenBox* (Ours)** | LiDAR + Camera | **62.74** / **28.03** | **51.55** / **15.06** | **20.08** / 1.88 |
| DBSCAN [7] | LiDAR | 1.94 / 0.25 | 0.19 / 0.00 | 0.20 / 0.00 |
| MODEST [43] | LiDAR | 15.83 / 5.48 | 8.96 / 0.10 | 0.43 / 0.20 |
| OYSTER [45] | LiDAR | 26.21 / 14.10 | 3.52 / 0.14 | 1.24 / 0.32 |
| CPD [40] | LiDAR | 50.18 / 32.13 | 18.01 / 13.22 | 5.61 / 4.87 |
| **OpenBox† (Ours)** | LiDAR + Camera | **58.42** / 33.72 | **47.78** / **31.77** | **20.19** / **6.81** |
| **OpenBox‡ (Ours)** | LiDAR + Camera | 51.70 / **34.95** | 33.02 / 23.50 | 7.95 / 5.99 |

