# OpenReview forum: "OpenBox: Annotate Any Bounding Boxes in 3D"
_NeurIPS.cc/2025/Conference — NeurIPS 2025 spotlight_

### Official Review · Reviewer_GKxa · 2025-06-26

**Clarity:** 3
**Significance:** 3
**Originality:** 3
**Rating:** 5
**Confidence:** 5

**Summary:**

This paper proposes OpenBox, a automatic 3D bounding box annotation pipeline that leverages 2D visual foundation models to generate high-quality 3D annotations for various object classes without requiring self-training. It achieves state-of-the-art performance on datasets like Waymo and Lyft by refining point clouds and adapting bounding boxes based on object physical properties.  Overall, the  paper is well-written, and the designs of 3D-2D IoU alignment and Common Statistics from ChatGPT are interesting and novel in the this filed.

**Questions:**

Please see Weaknesses.

**Ethical Concerns:**

["NO or VERY MINOR ethics concerns only"]

**Limitations:**

yes

**Paper Formatting Concerns:**

No formatting concern.

**Quality:**

3

**Strengths And Weaknesses:**

Strengths：

1 The paper is well-written, with clear methodology and extensive experimental results that validate its effectiveness.

2 The use of 3D-2D IoU alignment and Common Statistics from ChatGPT is interesting and novel in the this filed.

3 The motivation for the module design is quite clear, and the Figures are clear. For instance, by referring to Figures 2 and 3, one can quickly understand the entire process of the method and the details of the design.

Weaknesses:

1.	Lack of comparison with other image-involved methods.
I noticed that the paper only compares with LiSe. However, there are several other methods that also use image data for annotation, such as LSMOL [1] and UNION [2]. A more comprehensive comparison with these approaches would strengthen the evaluation.

2.	Missing discussion on related 2D-to-3D annotation works.
Some existing methods utilize 2D foundation models to generate 3D annotations, such as SP3D [3]. Although their problem settings may differ, they share similarities in generating 3D boxes from 2D cues. Before introducing the Context-aware Refinement module, it would be helpful to discuss how previous work has addressed the challenge of inaccurate unprojection.

3.	Clarification needed on size prior from ChatGPT.
The size estimates obtained from ChatGPT are generally stochastic in nature—different prompts and query times may lead to varying results. Therefore, it is important to clarify the details of how the size prior was obtained, including prompt formulation and query settings.

[1] Wang, Yuqi, Yuntao Chen, and Zhao-Xiang Zhang. "4d unsupervised object discovery." Advances in Neural Information Processing Systems 35 (2022): 35563-35575.
[2] Lentsch, T., Caesar, H., & Gavrila, D. (2024). Union: Unsupervised 3d object detection using object appearance-based pseudo-classes. Advances in Neural Information Processing Systems, 37, 22028-22046.
[3] Zhao, Shijia, Qiming Xia, Xusheng Guo, Pufan Zou, Maoji Zheng, Hai Wu, Chenglu Wen, and Cheng Wang. "SP3D: Boosting Sparsely-Supervised 3D Object Detection via Accurate Cross-Modal Semantic Prompts." arXiv preprint arXiv:2503.06467 (2025).

---

> ### Author Rebuttal · Authors · 2025-07-31
>
> We sincerely thank the reviewer for taking the time to provide such a valuable review. We appreciate the reviewer’s recognition of our paper’s well-written presentation, clear methodology, and extensive experiments that validate its effectiveness. The novel use of 3D-2D IoU alignment and ChatGPT-derived common statistics, as well as the clearly motivated module design and illustrative figures were particularly encouraging highlights. The reviewer’s insights will guide us in further refining both the technical depth and clarity of our work.
> Below, we provide detailed responses to the reviewer’s questions.
>
>
> **Q1. Lack of comparison with other image-involved methods. I noticed that the paper only compares with LiSe. However, there are several other methods that also use image data for annotation, such as LSMOL [1] and UNION [2]. A more comprehensive comparison with these approaches would strengthen the evaluation.**
>
>
> Table 1. 3D object detection results on nuScenes[3] validation set. All methods use CenterPoint[7] as a 3D object detector. L and C denote LiDAR and Camera respectively.
> | Method   | Modality | Car 3D AP ↑     | Pedestrian 3D AP ↑ | Cyclist 3D AP ↑ |
> |----------|----------|------------------|---------------------|------------------|
> | UNION    | L + C    | 30.1             | 41.6                | 0.0              |
> | **OpenBox (Ours)**  | L + C    | **40.9 (+10.8)**     | **62.7 (+21.1)**        |  **5.2 (+5.2)**       |
>
>
> As shown in Table 1, we observe performance improvements across all classes. One key reason for the significant gains is that, unlike OpenBox, UNION [2] omits the refinement process for point clouds and 3D bounding boxes. In particular, it does not explicitly consider the camera-lidar calibration error when projecting LiDAR point clouds on DINOv2 [4] feature maps which leads to noise at the boundary of the objects.. Additionally, UNION [2] neither resizes nor re-localizes the initial 3D bounding boxes, which leads the model to predict suboptimal bounding boxes.
>
>
> Table 2. 3D object detection results on Waymo Open Dataset [5] validation set. All values are measured under IoU = 0.4 (Intersection over Union). VRU indicates vulnerable road users (pedestrian + cyclist).
>
> | Method   | Modality | Vehicle 3D AP ↑      | VRU 3D AP ↑         |
> |----------|----------|------------------|---------------------|
> | UP-VL [6]| L + C    | 52.0             | 19.7                |
> | **OpenBox (Ours)**  | L + C    | **60.7 (+8.7)**      | **22.7 (+3.0)**         |
>
>
> Similarly, in Table 2, we observe a significant performance improvement compared to the previous baseline, UP-VL [6]. Although we could not verify this directly due to the lack of released code, Figure 6 (Error Analysis) in the supplementary material of UP-VL [6] shows that classification errors (confusion with background) account for a large portion of the mistakes. We conjecture that this is because UP-VL [6] applies feature distillation directly to the point clouds without any refinement process—unlike OpenBox, which employs context-aware refinement to mitigate noise introduced during 2D-to-3D unprojection. Additionally, while LSMOL [1] is a relevant recent work, it does not evaluate 3D object detection and thus was not included in our experiments. However, to further promote generalizable 3D perception, we plan to evaluate our method on the 3D instance segmentation task in future work. We will include these results and discussion in the final version upon acceptance.
>
> **Q2. Missing discussion on related 2D-to-3D annotation works. Some existing methods utilize 2D foundation models to generate 3D annotations, such as SP3D [6]. Although their problem settings may differ, they share similarities in generating 3D boxes from 2D cues. Before introducing the Context-aware Refinement module, it would be helpful to discuss how previous work has addressed the challenge of inaccurate unprojection.**
>
>
> In general, the 2D-to-3D un-projection process introduces significant noise, particularly around the boundaries of the 2D masks due to camera LiDAR calibration errors. To mitigate this, methods such as OVM3D-Det [9] and SP3D [8] attempt to reduce noise by shrinking the masks. However, these approaches still fail to address errors stemming from the imperfections of 2D visual foundation models. For example, when a vehicle is reflected on a storefront window, such reflections may be incorrectly projected into 3D space, resulting in false positives. In contrast, our context-aware refinement evaluates the ratio between clusters near reflective surfaces and the actual unprojected point cloud, allowing it to effectively remove such false positives.(Please see Table 4-(a) and Figure 4  in the main paper.)
>
> **Q3. Clarification needed on size prior from ChatGPT. The size estimates obtained from ChatGPT are generally stochastic in nature—different prompts and query times may lead to varying results. Therefore, it is important to clarify the details of how the size prior was obtained, including prompt formulation and query settings**
>
> To obtain 3D bounding box size priors and determine the rigidity of objects, we utilized the following prompts with ChatGPT-4 [10]:
>
> **Prompt**: Please provide the typical 3D bounding box size for *[class]*
>
> **Response**: Here are the typical 3D bounding box dimensions (Length × Width × Height in meters) for *[class]* , based on common datasets like nuScenes, KITTI, and the Waymo Open Dataset.
>
>
> **Prompt**: Is *[class]* deformable or rigid?
>
> **Response**: A *[class]* is considered a deformable / rigid object.
>
>
> Below is the list of categories we provided to Grounding DINO [11] as text prompts: \
> [Car, Bus, Person, Truck, Construction Vehicle, Trailer, Barrier, Bicycle, Motorcycle, Traffic Cone, Dog, Fire Hydrant, Stroller].
>
>
> [1] Wang et al. (2022). 4d unsupervised object discovery. In NeurIPS. \
> [2] Lentsch et al. (2024). UNION: Unsupervised 3D Object Detection using Object Appearance-based Pseudo-Classes. In NeurIPS. \
> [3] Caesar et al. (2020). nuscenes: A multimodal dataset for autonomous driving. In CVPR. \
> [4] Oquab et al. (2023). DINOv2: Learning Robust Visual Features without Supervision. arXiv. \
> [5] Sun et al. (2020). Scalability in perception for autonomous driving: Waymo open dataset. In CVPR. \
> [6] Najibi et al. (2023). Unsupervised 3D Perception with 2D Vision-Language Distillation for Autonomous Driving. In ICCV. \
> [7] Yin et al. (2021). Center-based 3D Object Detection and Tracking. In CVPR. \
> [8] Zhao et al. (2025). SP3D: Boosting Sparsely-Supervised 3D Object Detection via Accurate
> Cross-Modal Semantic Prompts. In CVPR. \
> [9] Huang et al. (2024). Training an Open-Vocabulary Monocular 3D Object Detection Model without 3D Data. In NeurIPS. \
> [10] OpenAI. (2023). GPT-4 Technical Report. arXiv. \
> [11] Liu et al. (2023). Grounding dino: Marrying dino with grounded pre-training for open-set object detection. arXiv.

---

> ### Author Response · Authors · 2025-08-08
> **Thank you for your feedback**
>
> Dear Reviewer GKxa,
> We thank reviewer GKxa for the follow-up. The recommendations to add more comparisons with baselines, include the ChatGPT prompts/responses, and discussion with related work for context-aware refinement module will materially improve the paper. These changes will be reflected in the revised manuscript and supplement. We remain ready to address any additional points.

---

### Official Review · Reviewer_cFQr · 2025-06-28

**Clarity:** 2
**Significance:** 3
**Originality:** 2
**Rating:** 5
**Confidence:** 4

**Summary:**

The manuscript introduces a system for annotating open world 3D bounding boxes for autonomous car datasets with aligned image and point cloud data. The approach leverages 2D foundation models and aligns those outputs effectively with 3D point cloud information as well as common sense 3D object sizes. The approach seems to perform well quantitatively and qualitatively. Some of the long tail 3D bounding boxes detected by the system like strollers and dogs are impressive.

**Questions:**

- see weaknesses;
    - It would also be very useful to see the baseline performance of the 3D OBB detector trained on the human annotated OBBs for both datasets.
    - clarify if code is going to be released
    - how is the mesh extracted from the point cloud?

**Ethical Concerns:**

["NO or VERY MINOR ethics concerns only"]

**Final Justification:**

I have read the rebuttal and the other reviews.

The authors have addressed my questions sufficiently. The additional baseline results with the same modalities support the contribution of the work. The eval against training straight on human GT is important to anchor the machine generated annotations. These and the interpretations should be added to the publication. I also appreciate the additional information on running time of the annotation tool as well as GPT prompts as suggested by other reviewers.

Trusting that authors will add this information to the publication I will raise my rating to accept.

**Limitations:**

limitations are in the main paper

**Quality:**

3

**Strengths And Weaknesses:**

- strength:
    - The key strength of the paper is the strong performance of the approach. The qualitative results look very good and really help drive home the quantitative results. The open world 3D OBBs like dog, stroller are great to see (although it is unclear how cherry picked they are?). The supplemental video does a great job at illustrating the performance of the method on dynamic scenes with many humans and cars and gives further evidence for the long-tail localization ability. The temporal stability of the 3D OBBs is also great to see.
    - The manuscript has good illustrations to help communicate the system components and interplay as well as describe details like 3D bounding box extension.
- weaknesses:
    - The quantitative evaluation only shows comparison to methods that leverage only LiDAR data on WOD except for LiSe on Lyft. The approach's performance would be stronger supported with baselines that leverage the same modalities on WOD as well. It would also be very useful to see the baseline performance of the 3D OBB detector trained on the human annotated OBBs for tboth datasets.
    - clarity and reproducability: The approach is a complex system of 2D foundation models, 3D clustering, 3D ground plane estimation, surface estimation, bounding box priors from ChatGPT. Even though the description in the paper is quite exhaustive and there are multiple good figures illustrating the system, I doubt that it can be reproduced well without release of the code (which is indicated for release in the paper checklist but not the main paper)
    - clarity: the writing clarity and quality could be improved.
        - For example, various acronyms are used without introduction which makes it hard to follow the paper. I.e. PP score (l144), CST, CBR, CPD (table 1, l218).
        - Another example is that it is not that clearly written that all the related work shown in tables are also annotating 3D OBBs and the numbers are from using the same model to train on the annotated datasets.
        - It is unclear how the mesh is reconstructed from the point cloud.

All in all the method seems to work really well but the complexity of the approach and some clarity issues in the writing will make it hard to reproduce. It will be important to release code to make this reproducible.

---

> ### Author Rebuttal · Authors · 2025-07-31
>
> Thank you for the encouraging feedback. We’re glad the qualitative results and supplemental video clearly illustrate our quantitative improvements and that the temporal stability and system illustrations were helpful. The reviewer’s insights will guide us in highlighting these strengths even more effectively.
> Below, we provide detailed responses to the reviewer’s questions. We will include additional results and discussion in the final version.
>
> ## Comparison with more baselines ##
>
> **Table 1.** 3D object detection results on nuScenes [1] validation set. All methods use CenterPoint [2] . L and C denote LiDAR and camera respectively.
>
> | Method    | Modality | Car 3D AP ↑    | Pedestrian 3D AP ↑ | Cyclist 3D AP ↑ |
> |:-----------|:----------:|:----------------|:---------------------|:------------------|
> | UNION [3] | L + C    | 30.1           | 41.6                | 0.0              |
> | **OpenBox**   | L + C    | **40.9 (+10.8)**   | **62.7 (+21.1)**        | **5.2 (+5.2)**       |
>
> As shown in Table 1, we observe performance improvements across all classes. One key reason for the significant gains is that, unlike OpenBox, UNION [3] omits the refinement process for point clouds and 3D bounding boxes. In particular, it does not explicitly consider the camera-LiDAR calibration error when projecting LiDAR point clouds on DINOv2 [4] feature maps which leads to noise at the boundary of the objects. Additionally, UNION [3] neither resizes nor re-localizes the initial 3D bounding boxes, which leads the model to predict suboptimal bounding boxes.
>
> **Table 2.** 3D object detection results on WOD [5] validation set. All values are measured under IoU=0.4  (Intersection of Union). VRU indicates vulnerable road users (pedestrian + cyclist).
>
> | Method    | Modality | Vehicle 3D AP ↑ | VRU 3D AP ↑     |
> |-----------|:----------:|:------------------|:------------------|
> | UP‑VL [6] | L + C    | 52.0             | 19.7             |
> | **OpenBox**   | L + C    |**60.7 (+8.7)**      | **22.7 (+3.0)**      |
>
> Similarly, in Table 2, we observe a significant performance improvement compared to the previous baseline, UP-VL [6]. Although we could not verify this directly due to the lack of released code, Figure 6 (Error Analysis) in the Supplementary Material of [6] shows that classification errors (confusion with background) account for a large portion of the mistakes. We conjecture that this is because UP-VL [6] applies feature distillation directly to the point clouds without any refinement process unlike ours which employs context-aware refinement to mitigate noise introduced during 2D-to-3D unprojection.
>
> ## Comparison with model trained on human annotated dataset. ##
>
> **Table 3.** 3D object detection results on WOD [5] validation set.
>
> | Annotation Method | Veh 3D AP (0.5 / 0.7) | Pedestrian 3D AP (0.3 / 0.5) ↑ | Cyclist 3D AP (0.3 / 0.5) ↑ |
> |:-------------------|:------------------------:|:-------------------------------:|:----------------------------:|
> | Human             | 93.31 / 75.70 | 87.25 / 77.93 | 58.33 / 54.88 |
> | Auto (Ours)       | 66.89 / 39.14 | 55.71 / 37.82 | 21.00 /   &nbsp;7.08 |
>
>
> **Table 4.** 3D object detection results on Lyft [7] validation set. Following [18], we evaluate in a class-agnostic manner at IoU = 0.25, and each value represents BEV AP / 3D AP.
>
>
> | Annotation Method | 0-30m         | 30-50m        | 50-80m        | 0-80m         |
> |:-------------------|:---------------:|:---------------:|:---------------:|:---------------:|
> | Human             | 82.8 / 82.6   | 70.8 / 70.3   | 50.2 / 49.6   | 69.5 / 69.1   |
> | Auto (Ours)       | 62.4 / 62.3   | 56.6 / 50.6   | 19.9 / 19.6   | 42.5 / 42.0   |
>
> Table 3 and Table 4 use [8], [9] for  3D object detector, respectively.  In Table 3, while all classes show a performance gap compared to models trained on human-annotated data, the cyclist class suffers the largest drop. This stems from limitations of 2D visual foundation models such as Grounding DINO [10]. Because [10] was not trained on the prompt “cyclist,” it cannot directly infer that category. We therefore substitute the prompt “bicycle,” which leads to inconsistent detections: in the same scene, the model may first detect both bicycle and rider, then detect only the bicycle **(please watch 01:09  in the supplementary video for a mis-localized cyclist)**. In Table 4, performance between 50 m and 80 m declines sharply. Even after aggregation, distant objects yield sparse point clouds that are then discarded during HDBSCAN [12] or subsequent refinement, leaving few annotated boxes in long range. Although our automatic annotations still underperform human labels, we expect that integrating a tracking module and dense, image-based 3D bounding box annotation [13] will help recover performance for both distant and challenging instances. Our method shows substantial progress toward human-level performance, producing training data of sufficient quality to achieve object detection results that are reasonably comparable to those trained on human-annotated datasets.
>
> ## Clarity of Methodology ##
>
> **Table 5.** Method and 3D object detection network used for main paper
>
> | Method Used on Dataset        | 3D Object Detection Network |
> |------------------------------|-----------------------------|
> | WOD [5]       | Voxel RCNN [8]              |
> | └ DBSCAN [14], MODEST [15], CPD [16], OYSTER [17] |                             |
> | Lyft [7]             | Point RCNN [9]              |
> | └ LiSe [18], MODEST [15]     |                             |
>
> Though we wrote the experimental settings in line 204 - line 207 in the main paper, we will include Table 5 for improving clarity of the paper. Additionally we will include the explanation of acronyms before methodology part in the main paper for readers who are not much familiar with this area. Below is the explanation of acronyms.
>
> PP Score (Persistence Point Score) [19] : Measurement of how “persistent” a LiDAR point is across multiple passes (traversals) of the same route. Higher means high probability of static point.
>
> CBR (CProto-constrained Box Regularization for Label Refinement)  [16]: CPD [16], an unsupervised 3D object detection method, computes a quality score for each initial pseudo label based on distance, occupancy, and size similarity. The high score pseudo labels (i.e., those of highest quality) are collected into a prototype set (CProto). These prototypes are then used to refine the original pseudo labels.
>
> CST (CProto-constrained Self-Training)  [16] : [16] proposes an optimization strategy for 3D object detection networks in which both the near, dense point clouds from the CProto set and the downsampled sparse point clouds are fed as inputs. The network’s predictions on these two inputs are then constrained via a feature consistency loss and a bounding-box regression loss.
> Mesh Reconstruction : We utilize VDBFusion [20] on a sequence of point clouds for mesh reconstruction.
>
> ## Other Questions ##
>
> We appreciate the reviewer's acknowledgement regarding the efficiency of our method. We will make sure to further clarify the paper in the final version of our submission. We are open to open source the code upon acceptance of our work. The title of our approach "OpenBox" aims at reflecting the duality between open-vocabulary and open source. We also include a list of github repositories that helped us construct our codebase[16,21,22,23].
>
> [1] nuscenes: A multimodal dataset for autonomous driving. In CVPR'20
> [2] Center-based 3D Object Detection and Tracking. In CVPR'21
> [3] UNION: Unsupervised 3D Object Detection using Object Appearance-based Pseudo-Classes. In NeurIPS'24
> [4] DINOv2: Learning Robust Visual Features without Supervision. arXiv'23
> [5] Scalability in perception for autonomous driving: Waymo open dataset. In CVPR'20.
> [6] Unsupervised 3D Perception with 2D Vision-Language Distillation for Autonomous Driving. In ICCV'23
> [7] One thousand and one hours: Self-driving motion prediction dataset. In CoRL'21
> [8] Voxel r-cnn: Towards high performance voxel-based 3d object detection. In AAAI'21.
> [9] PointRCNN: 3D Object Proposal Generation and Detection from Point Cloud. In CVPR'19.
> [10] Grounding dino: Marrying dino with grounded pre-training for open-set object detection. arXiv'23.
> [11] Sam 2: Segment anything in images and videos. arXiv'24.
> [12] A Hybrid Approach To Hierarchical Density-based Cluster Selection. In IEEE International Conference on Multisensor Fusion and Integration for Intelligent Systems, 2019 (MFI)
> [13] Training an Open-Vocabulary Monocular 3D Object Detection Model without 3D Data. In NeurIPS'24
> [14] A Density-Based Algorithm for Discovering Clusters in Large Spatial Databases with Noise. In Proceedings of 2nd International Conference on Knowledge Discovery and Data Mining (1996).
> [15] Learning to detect mobile objects from lidar scans without labels. In CVPR'22.
> [16] Commonsense prototype for outdoor unsupervised 3d object detection. In CVPR'24.
> [17] Oyster: Towards Unsupervised Object Detection from LiDAR Point Clouds. In CVPR'23
> [18] Approaching Outside: Scaling Unsupervised 3D Object Detection from 2D Scene. In ECCV'24.
> [19] Learning to detect mobile objects from lidar scans without labels. In CVPR'24.
> [20] VDBFusion: Flexible and Efficient TSDF Integration of Range Sensor Data. Sensors. 2022
> [21] OpenPCDet Development Team.  Openpcdet: An open-source toolbox for 3d object detection from point clouds. Github. 2020
> [22] Grounded SAM: Assembling Open-World Models for Diverse Visual Tasks. arXiv'24.
> [23] Patchwork++: Fast and robust ground segmentation solving partial under-segmentation using 3D point cloud. In IROS'22

---

> > ### Comment · Reviewer_cFQr · 2025-08-06
> >
> > I have read the rebuttal and the other reviews.
> >
> > The authors have addressed my questions sufficiently. The additional baseline results with the same modalities support the contribution of the work. The eval against training straight on human GT is important to anchor the machine generated annotations. These and the interpretations should be added to the publication.
> > I also appreciate the additional information on running time of the annotation tool as well as GPT prompts as suggested by other reviewers.
> >
> > Trusting that authors will add this information to the publication I will raise my rating to accept.

---

### Official Review · Reviewer_2oE1 · 2025-06-30

**Clarity:** 2
**Significance:** 3
**Originality:** 2
**Rating:** 4
**Confidence:** 3

**Summary:**

In this paper, author come up with a noval two stage automatic annotation pipeline for autonomous driving called OpenBox. In the first stage, Cross-modal Instance Alignment is executed. OpenBox first utilizes the instance masks generated by 2D visual model and unproject them back to 3D point cloud to generate instance level point cloud. The results are then fused with the result of point cloud clustering using HDBSCAN. The second stage is Adaptive 3D Bounding Box Generation, which generates boxes on the basis of the point cloud from stage1 and the physical types of each instance. Compared with other automatic annotation methods, OpenBox requires no self-training and generates more accurate annotation using context-aware refinement and surface-aware noise filtering. Experiment results on Waymo show OpenBox-generated annotations train the best performaned model achieving 70.49% AP3D for the vehicle class and results on Lyft shows better performance compared with model trained on human annotation boxes.

**Questions:**

## Questions:
Annotation cost is pointed out in the abstract, and one of the contribution of OpenBox is that it requires no training of 3D object detectors. Typically 3D bounding box is decided by its center, size and rotation. The localization(center) of bounding box is substituted by 2D segmentation models and 3D point cloud clustering. Box size follows the typical size of the category and rotation is decided by 3D-2D IoU alignment. It is great to combine existing methods and provide amazing annotation effects. However shall we neglact the training cost of the existing methods like DINO, SAM2 and ChatGPT. Furthermore, what is the inference cost of all models. Are all the models/algorithms used in OpenBox use less resources/cost less time than a traditional 3D object detection? If possible, please provide these information in the appendix and we can have a better comparison between OpenBox and traditional 3D object detection way.

## Suggestions:
1. The author should provide the missing experiment settings and appendix.
2. More details about the auto-annotation process are welcomed, like the inference cost(time/computation) of the methods/models used and the template to and the response from ChatGPT.

**Ethical Concerns:**

["NO or VERY MINOR ethics concerns only"]

**Final Justification:**

The problems and weakness of the initial review is well explained in the rebuttal. And it is true we should collect and use the existing fundamental models rather than 'build new wheels'. Since the author proves the inference cost of OpenBox is same or less than traditional 3D object detection models, I have no major questions about the paper and improve the Clarity score from 1 to 2 and final score from 3 to 4.

**Limitations:**

yes

**Quality:**

3

**Strengths And Weaknesses:**

Strengths:
- The experiment on WayMo and Lyft shows convincing evidence to the ability of OpenBox. With no more training required, OpenBox combines existing models (Grounding DINO, SAM2, ChatGPT) and point cloud processing methods(SDF, HDBSCAN) with noval postprocess on static, dynamic and deformable categories, providing a economical pipeline to perform high-quality anto-annotation task, helping improve the 3D object detection performance in autonomous driving.

Weaknesses:
- The paper fails to provide the detailed of Setting 1 and Setting 2 in section Experiments, which makes the whole section confusing.
- The paper lacks details of some process. For example, in section 3.2 ChatGPT is used twice to decide the type of the object and retrieval the typical size of the object. It's better to provide some conversation examples.
- The paper mentions about the appendix in Line 209 but no appendix is found.
- The paper fails to provide a code to reproduce the experiment results in the paper

---

> ### Author Rebuttal · Authors · 2025-07-31
>
> We appreciate the reviewer's recognition of our results on Waymo and Lyft. As noted, OpenBox requires no additional training yet seamlessly integrates existing models (Grounding DINO, SAM2, ChatGPT) with point cloud techniques (SDF, HDBSCAN) and a novel postprocess for static, dynamic and deformable categories. This cost-efficient pipeline delivers high-quality auto-annotations and drives improvements in 3D object detection for autonomous driving.
>
> Below, we provide detailed responses to the reviewer's questions.
>
> ## Details about the automatic annotation process ##
>
> **ChatGPT prompt**
>
> To obtain 3D bounding box size priors and determine the rigidity of objects, we utilized the following prompts with ChatGPT-4 [1]:
>
> - **Prompt:** Please provide the typical 3D bounding box size for *[class]*
>
>     **Response:** Here are the typical 3D bounding box dimensions (Length × Width × Height in meters) for *[class]*, based on common datasets like nuScenes, KITTI, and the Waymo Open Dataset.
>
> - **Prompt:** Is *[class]* deformable or rigid?
>
>     **Response:** A *[class]* is considered a deformable / rigid object.
>
> Below is the list of categories we provided to GroundingDINO [2] as text prompts:
> [Car, Bus, Person, Truck, Construction Vehicle, Trailer, Barrier, Bicycle, Motorcycle, Traffic Cone, Dog, Fire Hydrant, Stroller].
>
>
> ## Time/Computation cost for automatic annotation by ours ##
>
> We use 8 NVIDIA A6000 GPUs (48GB) and 2 AMD EPYC 7763 CPUs. Below is a summary of the datasets used in our experiments:
>
> - **nuScenes Dataset [3]:** approximately 150 K images and 28 K LiDAR scans
> - **Waymo Open Dataset [4]:** approximately 750 K frames images and 150K LiDAR scans
> - **Lyft Dataset [5]:** approximately 120 K images and 20 K LiDAR scans
>
> **Table 1**. Time cost for automatic annotation on various autonomous driving datasets.
>
> | | nuScenes [3] | Waymo [4] | Lyft [5] |
> |:---|---:|---:|---:|
> | Instance-level Feature Extraction | ~13 hours | ~1 day | ~10 hours |
> | Context Aware Refinement |~ 8 hours | ~10 hours | ~8 hours |
> | Adaptive 3D Bounding Box Generation | ~5 hours | ~12 hours | ~4 hours |
>
> **Table 2**. Time cost comparison with baselines. We utilize UNION [12] and CPD [6] as a baseline for nuScenes [3] and Waymo dataset [4] respectively.
>
> | | nuScenes [3] | Waymo [4] |
> |:---:|---:|---:|
> | **Ours** | ~26 hours | ~46 hours |
> | Baseline | ~46 hours | ~48 hours |
>
> Table 1 presents the time required for 3D bounding box annotation using OpenBox across different datasets. The relatively higher annotation time for the Waymo dataset [4] is attributed to its denser point clouds (64 channels) and larger dataset size. Table 2 compares the automatic annotation cost of OpenBox with other baselines. Notably, in the case of nuScenes [3], OpenBox achieves better performance despite requiring approximately 1.7× less annotation time **(please see Table 1 in the response for Reviewer GKxa)**. Similarly, for the Waymo dataset [4], although the annotation time is comparable, training with data annotated by OpenBox leads to superior detection performance (please see Table 1 in the main paper).
>
> ## Training time for 3D object detection model ##
>
> - Voxel-RCNN on Waymo Open Dataset: 18 hours
> - Point RCNN on Lyft dataset: 5 hours
> - Centerpoint on nuScenes dataset: 5 hours
>
> ## Clarity ##
>
> To comprehensively evaluate the effectiveness of OpenBox, we conduct experiments under two different settings. We will include the information below in the main paper.
>
> - **Setting 1:** The training set is generated via automatic annotation, which is then used to train a 3D object detector. Evaluation is performed on a human-annotated validation set.
>
> - **Setting 2:** We directly compare automatically annotated data with human-annotated data on training set.
>
> In the main paper, Table 1 and Table 2 correspond to Setting 1, while Table 3 and Table 4 present results from Setting 2. In addition, in Table 1, the symbol † indicates that we flip the OpenBox annotations and point clouds to obtain 360° coverage. The symbol ‡ denotes that the region outside the camera frustum is filled with annotations from CPD [6].
>
> "Appendix" refers to the supplementary material. Apologies for the confusion. Please refer to the supplementary material for additional information.
>
> ## Other Questions ##
>
> We appreciate the reviewer's acknowledgement regarding the efficiency of our method. We will make sure to further clarify the paper in the final version of our submission. We are open to open source the code upon acceptance of our work. The title of our approach "OpenBox" aims at reflecting the duality between open-vocabulary and open source. We also include a list of github repositories that helped us construct our codebase: OpenPCDet [7], CPD [6], GroundedSAM2 [8], pypatchwork++ [9]
>
> **Q: Annotation cost is pointed out in the abstract, and one of the contribution of OpenBox is that it requires no training of 3D object detectors. Typically 3D bounding box is decided by its center, size and rotation. The localization(center) of bounding box is substituted by 2D segmentation models and 3D point cloud clustering. Box size follows the typical size of the category and rotation is decided by 3D-2D IoU alignment. It is great to combine existing methods and provide amazing annotation effects. However shall we neglect the training cost of the existing methods like DINO, SAM2 and ChatGPT. Furthermore, what is the inference cost of all models. Are all the models/algorithms used in OpenBox use less resources/cost less time than a traditional 3D object detection? If possible, please provide these information in the appendix and we can have a better comparison between OpenBox and traditional 3D object detection way.**
>
> I agree to some extent with the reviewer's point that the training costs of Grounding DINO, SAM2, and ChatGPT cannot be ignored. However, all these methods are evolving into foundation models in a broader sense, serving vision tasks much like an ImageNet [10] pretrained ResNet [11] backbone. Indeed, our motivation is to leverage human level 2D visual foundation models to automate the dataset generation required for training a 3D foundation model.
>
> [1] OpenAI. (2023). GPT-4 Technical Report. arXiv.
> [2] Liu et al. (2023). Grounding dino: Marrying dino with grounded pre-training for open-set object detection. arXiv.
> [3] Caesar et al. (2020). nuscenes: A multimodal dataset for autonomous driving. In CVPR.
> [4] Sun et al. (2020). Scalability in perception for autonomous driving: Waymo open dataset. In CVPR.
> [5] Houston et al. (2021). One thousand and one hours: Self-driving motion prediction dataset. In Conference on Robot Learning.
> [6] Wu et al. (2024). Commonsense prototype for outdoor unsupervised 3d object detection. In CVPR.
> [7] OpenPCDet Development Team. (2020). Openpcdet: An open-source toolbox for 3d object detection from point clouds. Github.
> [8] Ren et al. (2024). Grounded SAM: Assembling Open-World Models for Diverse Visual Tasks. arXiv.
> [9] Lee et al. (2022). Patchwork++: Fast and robust ground segmentation solving partial under-segmentation using 3D point cloud. In IROS.
> [10] Deng et al. (2009). ImageNet: A large-scale hierarchical image database. In CVPR.
> [11] He et al. (2015). Deep Residual Learning for Image Recognition. In CVPR.
> [12] Lentsch et al. (2024). UNION: Unsupervised 3D Object Detection using Object Appearance-based Pseudo-Classes. In NeurIPS.

---

> > ### Comment · Reviewer_2oE1 · 2025-08-05
> >
> > Detailed inference costs are demonstrated in the rebuttal and OpenBox does provides a better performance with the same or less resources. These statistics as well as the GPT prompts/responses should all be placed in the paper to improve the clarience of the paper.

---

> ### Author Response · Authors · 2025-08-08
> **Thank you for your feedback**
>
> We thank reviewer 2oE1 for engaging with the rebuttal and for the concrete suggestions. We will incorporate per-stage costs for automatic annotation and include the exact ChatGPT prompts and representative responses in the main paper and supplement to improve clarity. We greatly appreciate an increased score!

---

### Official Review · Reviewer_WBAe · 2025-07-01

**Clarity:** 3
**Significance:** 3
**Originality:** 3
**Rating:** 4
**Confidence:** 4

**Summary:**

This work proposes an auto-labeling pipeline for annotating 3D bounding boxes by leveraging 2D visual foundation models. The pipeline first segments object instances using visual knowledge distilled from foundation models and aggregated multi-frame LiDAR point clouds. It then generates geometrically adaptive bounding boxes by incorporating category-specific prior knowledge, such as dynamic/non-rigid properties and real-world size distributions. The proposed approach produces high-fidelity annotations without requiring iterative self-training, demonstrating significant potential for scalable 3D dataset generation.

**Questions:**

1. Are there alternative methods for refining the bounding box fit to the segmented instance point clouds? Additionally, what is the quantitative impact on detection precision (e.g., mAP, IoU) if bounding box fitting is omitted from the pipeline?
2. Which object categories present the most significant challenges for accurate estimation within this pipeline? Furthermore, could representative failure cases or limitations be discussed to provide deeper insight.

**Ethical Concerns:**

["NO or VERY MINOR ethics concerns only"]

**Final Justification:**

The rebuttal effectively addresses my concerns. Although the techniques used in this work are not novel, it demonstrates significant practical value for industrial applications. I am inclined to accept this work and look forward to its open-sourcing.

**Limitations:**

Yes

**Paper Formatting Concerns:**

No formatting issues.

**Quality:**

3

**Strengths And Weaknesses:**

Strengths:
1.This auto-labeling framework has potential value on open-world perception as annotating 3D instance is very expensive.
2.The method accounts for diverse object properties, including rigid, non-rigid, and dynamic objects, making it applicable to a wide range of real-world scenarios.
3. The pipeline does not relying on self-training which makes it more practical for deployment.

Weakness:
1. Since the point clouds only encode the partial view of an object, how the bounding-boxes are adjusted to fit the instance. It seems that the most critical part is using PP Score and point-to-edge closeness for bounding-boxes adherence. It is better to illustrate more in case the reader do not familiar with the cited work.
2. The bounding box extension based on the normal and ray direction is not general. The deformation of real-world objects can be in any form.
3. The orientation estimation on the static objects remains unclear.

---

> ### Author Rebuttal · Authors · 2025-07-31
>
> We sincerely appreciate the reviewer's positive feedback on the novelty and practicality of our proposed framework. We are encouraged to see that the reviewer recognizes: (1) the value of automatic labeling for 3D perception, especially given the high cost of 3D instance annotation in open-world settings (2) the generality of our method, which explicitly handles diverse object types including rigid, non-rigid, and dynamic objects enabling wide applicability across real-world scenarios and (3) the fact that our pipeline does not rely on self-training, which improves its practicality and robustness for real-world deployment.
>
> **Q1. Are there alternative methods for refining the bounding box fit to the segmented instance point clouds? Additionally, what is the quantitative impact on detection precision (e.g., mAP, IoU) if bounding box fitting is omitted from the pipeline?**
>
> We agree with the reviewer’s concern that the bounding box extension based on surface normals and ray directions is not universally applicable. As noted in our main paper (lines 283–285), we acknowledge the limitations of our approach, especially when handling deformable classes. The primary reason we adopted this extension method is that we assume rigid objects, making this strategy suitable for the majority of cases. To address the reviewer’s concern and better handle segmented instance point clouds, we believe a more robust refinement could be achieved by first increasing point cloud density through instance-level registration, utilizing instance IDs obtained from SAM2 [3]. Then, using point-aware bounding box optimization with techniques such as Ray Tracing Loss [6] and Point Ratio Loss [6], along with the Closeness-to-Edge criterion [9], it would be possible to more accurately refine bounding boxes for deformed static objects. We consider this a promising direction for future work.
>
> Additionally, in response to the reviewer’s suggestion, we conducted an ablation study to assess the impact of the bounding box refinement module, as presented in Table 1.
>
> **Table 1**. Ablation study of 3D bounding box refinement module on WOD [1] validation set. V, P and C indicate Vehicle, Pedestrian and Cyclist. AP 3D (Average Precision for 3D) for each classes are evaluated under IoU=0.7/0.5/0.5 respectively.
> Visibility-based is designed for refining dynamic rigid objects, while the 3D-2D IoU alignment targets static rigid objects for bounding box refinement.
>
> | | Visibility-based | 3D-2D IoU alignment | AP 3D ↑(V/P/C) |
> |:---:|:---:|:---:|:---:|
> | (a) | | | &nbsp; 2.95 /  &nbsp; 0.00 / 0.00 |
> | (b) | | ✔ | 29.97 / 13.25 / 0.94 |
> | (c) | ✔ | | 26.33 / 12.82 / 1.08 |
> | (d) | ✔ | ✔ | 32.41 / 17.11 / 2.15 |
>
> Table 1 illustrates the impact of the 3D bounding box refinement module on detection performance. We use Voxel R-CNN [2] as the 3D object detector, which is trained on an automatically annotated version of the Waymo Open Dataset (WOD) [1] using OpenBox, and evaluated on the human-annotated WOD [1], following setting 1 described in the main paper. Both training and evaluation are performed within the camera frustum region. When comparing (a) and (d), we observe improvements in AP 3D performance across all classes. This is because (a) generates sub-optimal 3D bounding boxes in regions where the LiDAR point cloud captures partial views due to either long-range distances or occlusion. And it causes the model to predict boxes which fail to reach the required IoU thresholds with the GT boxes used for AP 3D evaluation. Furthermore, when examining (b), (c), and (a) collectively, we see that (b) (Rigid Static) shows a larger performance gain from (a) compared to (c) (Rigid Dynamic). This is attributed to the fact that parked vehicles are more prevalent than moving vehicles in autonomous driving datasets. Moreover, static instances undergo aggregation, resulting in denser point clouds, and bounding boxes are generated after two stages of point-level filtering (context-aware and surface-aware). In contrast, dynamic instances do not benefit from aggregation and are filtered only by the context-aware module, leading to higher uncertainty.
>
> **Q2. Which object categories present the most significant challenges for accurate estimation within this pipeline? Furthermore, could representative failure cases or limitations be discussed to provide deeper insight.**
>
> We regret that policy restrictions prevent us from including figures illustrating those failures and challenging cases. We will include the failure case in the supplementary material with explanation below.
>
> Reviewing the quantitative results, annotating cyclists with 3D bounding boxes proves most challenging. This stems from imperfections of 2D visual foundation models such as GroundingDINO [8] and SAM2 [3]. Since GroundingDINO's [8] training did not include the prompt "cyclist," it cannot infer that class. We therefore use the prompt "bicycle." In the same scene, the model may first detect both bicycle and rider together and then detect only the bicycle (This inconsistency is clear in the supplementary video at 01:09, where a cyclist is mislocalized) As a result, the SAM2 mask size changes and the unprojected point cloud varies. Even with a size prior, these variations prevent accurate 3D bounding box annotation.
>
> We identified three additional failure modes. First, distant objects sometimes are not detected by GroundingDINO [8], 3D bounding boxes cannot be generated **(01:53 \\(\\sim\\) 01:56 in the supplementary video, traffic-cone appears without consistency)**. Second, even after point cloud aggregation, distant objects remain too sparse and vanish during refinement. **(00:24 sec in the supplementary video, a distant static vehicle is omitted)**. Last, imperfections in the PP score [4] can misclassify dynamic objects as static, producing false positives. **(01:13 in the supplementary video, a false positive 3D box appears behind a purple vehicle annotation.)**
>
> To mitigate afore-mentioned issues are as follows. First, using detection by tracking algorithms such as AB3DMOT [5] could provide a degree of consistency. Second, fusing image based 3D auto annotation methods such as OVM-Det3D [6] could address sparsity. Finally, leveraging scene flow (VoteFlow[7]) to predict object motion more robustly is expected to substantially reduce false positives.
>
> ## Clarity ##
>
> We will include the explanation of some related works before methodology part in the main paper for readers who are not much familiar with this area. Below is the explanation of PPscore [4] and point-to-edge closeness.
>
> **PP Score (Persistence Point Score) [4]:** Measurement of how "persistent" a LiDAR point is across multiple passes (traversals) of the same route. Higher means high probability of static point.
>
> **point-to-edge closeness [9]:** The closeness score measures how well a candidate rectangle hugs a point set by summing each point's inverse distance to its nearest rectangle edge. Rectangles with more points lying close to their sides get higher scores.
>
> To determine the static rigid box orientation using the closeness criterion, we proceed as follows:
>
> 1. **Sample candidate orientations:** We rotate the point set by a series of angles (e.g. 0°, 10°, 20°, … up to 180°).
>
> 2. **Evaluate snugness at each angle:** For each rotated set, we fit the smallest axis-aligned rectangle around the points and compute the closeness score by summing each point's inverse distance to its nearest rectangle edge.
>
> 3. **Select the optimal orientation:** The angle that yields the highest closeness score is chosen as the final orientation, since it indicates the rectangle that "hugs" the point cloud most tightly.
>
> This procedure reliably identifies the best alignment by directly measuring how well the rectangle encloses the observed points.
>
>
>
> [1] Sun et al. (2020). Scalability in perception for autonomous driving: Waymo open dataset. In CVPR.
> [2] Deng et al. (2021). Voxel R-CNN: Towards high performance voxel-based 3D object detection. In AAAI.
> [3] Ravi et al. (2024). SAM 2: Segment anything in images and videos. arXiv.
> [4] You et al. (2022). Learning to detect mobile objects from LiDAR scans without labels. In CVPR.
> [5] Weng et al. (2020). AB3DMOT: A baseline for 3D multi-object tracking and new evaluation metrics. In IROS.
> [6] Huang et al. (2024). Training an open-vocabulary monocular 3D object detection model without 3D data. In NeurIPS.
> [7] Lin et al. (2025). VoteFlow: Enforcing local rigidity in self-supervised scene flow. arXiv.
> [8] Liu et al. (2023). Grounding DINO: Marrying DINO with grounded pre-training for open-set object detection. arXiv.
> [9] Zhang et al. (2017). Efficient L-shape fitting for vehicle detection using laser scanners. In IEEE IV (Intelligent Vehicles Symposium).

---

### Decision · Program_Chairs · 2025-09-17

**Decision:**

Accept (spotlight)

**Comment:**

This work proposes a two-stage auto-labeling framework for annotating open world 3D bounding boxes by leveraging 2D visual foundation models.
It received two accept and two boardline accept.
Most of the concerns have been solved during the rebuttal.
The AC recommends acceptance for this paper.